# Pivotal Role of AKT2 during Dynamic Phenotypic Change of Breast Cancer Stem Cells

**DOI:** 10.3390/cancers11081058

**Published:** 2019-07-26

**Authors:** Petra Gener, Diana Rafael, Joaquin Seras-Franzoso, Anna Perez, Luis Alamo Pindado, Glòria Casas, Diego Arango, Yolanda Fernández, Zamira V. Díaz-Riascos, Ibane Abasolo, Simó Schwartz

**Affiliations:** 1Drug Delivery and Targeting Group, Molecular Biology and Biochemistry Research Centre for Nanomedicine (CIBBIM-Nanomedicine), Vall d’Hebron Institut de Recerca, Universitat Autònoma de Barcelona, 08035 Barcelona, Spain; 2Networking Research Centre for Bioengineering, Biomaterials, and Nanomedicine (CIBER-BBN), Instituto de Salud Carlos III, 28029 Madrid, Spain; 3Biomedical Research in Digestive Tract Tumors, CIBBIM-Nanomedicine, Vall d’Hebron Institut de Recerca, Universitat Autònoma de Barcelona, 08035 Barcelona, Spain; 4Functional Validation & Preclinical Research (FVPR), CIBBIM-Nanomedicine, Vall d’Hebron Institut de Recerca, Universitat Autònoma de Barcelona, 08035 Barcelona, Spain

**Keywords:** cancer stem cells (CSC), dynamic phenotype, epithelial-to-mesenchymal transition (EMT), AKT2 targeting

## Abstract

Therapeutic resistance seen in aggressive forms of breast cancer remains challenging for current treatments. More than half of the patients suffer from a disease relapse, most of them with distant metastases. Cancer maintenance, resistance to therapy, and metastatic disease seem to be sustained by the presence of cancer stem cells (CSC) within a tumor. The difficulty in targeting this subpopulation derives from their dynamic interconversion process, where CSC can differentiate to non-CSC, which in turn de-differentiate into cells with CSC properties. Using fluorescent CSC models driven by the expression of *ALDH1A 1*(aldehyde dehydrogenase 1A1), we confirmed this dynamic phenotypic change in MDA-MB-231 breast cancer cells and to identify Serine/Threonine Kinase 2 (AKT2) as an important player in the process. To confirm the central role of AKT2, we silenced AKT2 expression via small interfering RNA and using a chemical inhibitor (CCT128930), in both CSC and non-CSC from different cancer cell lines. Our results revealed that AKT2 inhibition effectively prevents non-CSC reversion through mesenchymal to epithelial transition, reducing invasion and colony formation ability of both, non-CSC and CSC. Further, AKT2 inhibition reduced CSC survival in low attachment conditions. Interestingly, in orthotopic tumor mouse models, high expression levels of AKT2 were detected in circulating tumor cells (CTC). These findings suggest AKT2 as a promising target for future anti-cancer therapies at three important levels: (i) Epithelial-to-mesenchymal transition (EMT) reversion and maintenance of CSC subpopulation in primary tumors, (ii) reduction of CTC and the likelihood of metastatic spread, and (iii) prevention of tumor recurrence through inhibition of CSC tumorigenic and metastatic potential.

## 1. Introduction

Drug resistance in breast cancer cell populations is acquired through phenotypic changes mediated by either epithelial-to-mesenchymal transition (EMT), or tumor cell drift towards a cancer stem cell (CSC) phenotype [1,2]. As a consequence of phenotypic dynamism and clonal evolution during the cancer treatment, tumors contain a heterogeneous mixture of various cell types, with different stem-like and mesenchymal phenotypes. Moreover, in this intricate network made of diverse cell populations, the characteristics of mesenchymal and stem-like phenotype are often intertwined, becoming difficult to distinguish from one another [2,3,4,5]. EMT cells, as well as CSC, are described as responsible for metastatic spread, tumor progression, and resistance. CSC also possesses a tumor initiating capacity and cell renewal capacity, even though some mesenchymal cells have been shown to be able to initiate tumor growth. Moreover, the EMT conversion stimulates expression of CSC genes and vice-versa, thus CSC often expresses genes characteristic for mesenchymal cells [1,2,6].

Due to the high level of complexity, the investigation of dynamic phenotypic changes during cancer development and/or treatment has proven technically challenging, and the use of in vitro or in vivo models for this purpose are fairly limited. Permanently tagged CSC are able to mimic, at least to some extent, the complexity of phenotypic dynamism in laboratory conditions. Previously, we described CSC tagging based on an expression vector containing the tdTomato fluorescent protein under the control of *ALDH1A1* (aldehyde dehydrogenase 1A1) promoter [7]. Since ALDH1A1 is expressed in CSC, this approach enables the identification and isolation of CSC, and has proved successful for the MCF7 breast cancer cell line and HCT116 colon cancer cell line [7]. Here, we use the same strategy to identify and trace CSC from the highly aggressive triple-negative breast cancer (TNBC) cell line, MDA-MB-231. TNBC presents the overall worst prognosis, greater metastatic potential, and higher relapse rate compared to other breast cancer types [8]. Sequential single-agent chemotherapy remains the standard of care for patients with metastatic TNBC, since targeted therapies have failed due to the lack of commonly targetable receptors (estrogen receptor (ER), progesterone receptor (PR), and HER2/neu). Consequently, overall survival among patients with this form of breast cancer has not changed over the past 20 years; this highlights the need for advances in therapeutic options for these patients [9,10]. Two PARP (poly ADP ribose polymerase) inhibitors (Olaparib, Talazoparib) for patients with *BRCA*-mutated tumors, and immunotherapy for patients with PD-L1 (programmed death-ligand 1)—positive TNBC have been recently approved, but longer follow-ups are needed to determine how these agents improve overall survival among patients with TNBC [11].

In this study, we explored the importance of the AKT2/TWIST and AKT2/mTOR axis for CSCsurvival and malignancy. Even though AKT2 has previously been identified as a promising target for cancer treatment, its role in determining CSC phenotype has not been addressed [12,13,14].

AKT2 was chosen for this study as the main subtype of the protein kinases B family. AKT2 has been previously shown to have a direct relationship with malignant transformation and tumor dissemination [15,16,17]. In contrast, cells transfected with AKT1 and AKT3 exhibit only minor alterations in cell invasiveness and these isoforms do not appear to play a role during EMT [18,19]. Early works have also demonstrated that siRNA (small interfering RNA) depletion of AKT2 results in reduced cell growth and invasiveness, with MET only occurring in cells overexpressing AKT2, confirming its potential role in tumorigenesis [20]. Even though *AKT2* is not frequently over-amplified in breast cancer, its expression is linked with poor prognosis. In contrast, no correlation has been detected with *AKT1* or *AKT3* in terms of patients’ survival [21,22].

AKT2 is a major downstream effector of the canonical PI3-K (phosphoinositide 3-kinase) pathway, which appears generally associated with acquisition of the malignant phenotype in cancer cells [17,18,19]. AKT/PKB is a key regulator of various cell processes and its signaling outcome depends on cellular background and context. Similarly, all cancer-related molecular mechanisms are highly dependent on cell type and phenotype. AKT2 may act via different signaling mechanisms; TWIST and mTOR being the main downstream effectors of AKT2 [15,20,21]. Within the context of CSC, TWIST (twist family bHLH transcription factor 1) arose as a very promising candidate due to its important and well-known role in tumor invasion, migration, dissemination, and drug resistance [23,24]. Since TWIST-mediated effects are regulated by AKT2, silencing of this oncogene could be explored as a potential strategy to reduce TWIST-mediated EMT through decrease of E-CADHERIN expression [20,25]. Furthermore, mTOR (mechanistic target of rapamycin kinase) signaling activation, as a result of increased activity of PI3K/AKT, significantly contributes to the initiation and development of tumors due to their involvement in cell growth, proliferation, motility, invasion, and survival. In this context, mTOR activity is found deregulated in many types of cancer including breast, prostate, lung, melanoma, bladder, brain, and renal carcinomas [26,27]. Moreover, recent data also suggest that the PI3K/AKT2/mTOR signaling pathway strongly modulates CSC biology [14,28,29,30]. Targeting the EMT signaling axis via AKT2/TWIST and PI3K/AKT2/mTOR in order to revert EMT and restore the epithelial phenotype appears to be a promising strategy in cancer therapy.

## 2. Results

### 2.1. Characterization and Dynamism of MDA-MB-231 CSC-Like Cell Model

In order to tag TNBC cells with a CSC phenotype, MDA-MB-231 cells were stably transfected with ALDH1A1-tdTomato reporter vector, and flow sorted based on tdTomato fluorescence (Figure 1A,B). Consecutive cell passages of tdTomato^+^ MDA-MB-231 cells led to an initial decline and then stabilization of the tdTomato^+^ subpopulation (Figure 1C), which accounted for 14.7% ± 2.8% of tdTomato^+^ cells within the MDA-MB-231-ALDH1A1/tdTomato cell line (Figure 1B). The stem like nature of tdTomato^+^ cells was confirmed by increased expression of stem cell markers compared to tdTomato^−^ cells (Figure 1D).

Briefly, the relative increase of *ALDH1A1* mRNA in tdTomato^+^ CSC was 4.46 ± 0.53 fold (*p* = 0.025). Other CSC markers, such *NOTCH4*, *POU5F1/OCT4*, *ALOX5*, *CMKLRI*, *ABCG2*, and *CXCR1* were also found over-expressed (1.78 ± 0.18, *p* = 0.01; 1.78 ± 0.09, *p* = 0.017; 2.42 ± 0.06, *p* = 0.04; 1.89 ± 0.12, *p* = 0.02, 1.99 ± 0.08, *p* = 0.04; 3.1 ± 0.78, *p* = 0.034; fold, respectively). No difference in *CD44* and *CD24* mRNA levels were observed in MDA-MB-231 tdTomato^+^ cells compared to tdTomato^−^ (non-CSC; Appendix A). Of note, overall expression of CD44 was high, and CD24 was low in MDA-MB-231 cells according to their mesenchymal-stem like (MSL) characteristics (Appendix A) [31].

As expected, tdTomato^+^ MDA-MB-231 CSC-like cells were able to grow as mammospheres in low attachment cell culture conditions, when seeded in a medium without serum (Figure 2).

Nevertheless, the tdTomato^−^ (non-CSC) cells were also able to grow in these conditions but formed significantly fewer mammospheres than equivalent amounts of tdTomato^+^ cells (Figure 2, Table 1). While tdTomato^+^ CSC-like cells start to form the mammospheres when just 10 cells/well were seeded (mammosphere incidence = 6/12; 50 (*n*/*n*, (%))), it was necessary to seed at least 100 cells/well of tdTomato^−^ non-CSC cells to induce mammosphere formation (mammosphere incidence = 1/8; 12 (*n*/*n*, (%)). In contrast, 12/12; 100 (*n*/*n*, (%)) of incidence was observed when 100 CSC-like cells were seeded per well (Table 1).

Interestingly, when only the tdTomato^−^ cell population was cultured in non-attachment conditions; tdTomato^+^ cells appeared from day 7 within the cell culture (Figure 3). This was not due to a fast proliferation of some residual CSC; since it was possible to observe a sudden switch to tdTomato^+^ phenotype of the whole mammosphere (Appendix A). These “new” tdTomato^+^ cells expressed a significantly higher amount of stem cell markers (*ALDH1A1*, *NOTCH4*, *POU5F1/OCT4*, *ALOX5*, and *CMKLR1*) and had increased invasion potential, compared to sorted tdTomato^−^ cells (Figure 3D,E). Interestingly, the increase of tdTomato and stem cell gene expression in originally tdTomato^−^ cells, was just transient, since when “new” tdTomato^+^ CSC were cultured back in attachment conditions, in complete medium (four days), the expression of tdTomato and stem cell markers decreased back to normal and just few tdTomato^+^ cells (5.6%) were detected by flow cytometry (Figure 3). Analogously, we observed de novo tdTomato expression and stem cell gene expression, as well as increased invasion capacity, when using tdTomato^−^ cells sorted from an aggressive HCT8 colon cancer cell line (Appendix A). Of note, tdTomato^−^ MCF7 epithelial cells did not survive in mammosphere culture conditions.

In order to determine the tumor initiating capacity of MDA-MB-231 tdTomato^+^ cells, sorted tdTomato^+^, and tdTomato^−^ cells were xenotransplanted into mice. Two groups (eight animals each) were injected with 1000 cells into their flanks, and tumors were allowed to grow for 71 days. Tumors were detected in 8/8 animals in the group of tdTomato^+^ CSC, as expected, while just in 6/8 animals that were xenotransplanted. In both cases, obtained tumor volumes were similar (Figure 4C). Ex-vivo analysis of tumors showed expression of tdTomato (59.8% ± 5.3%) in all tumors of the CSC-like group. However, the presence of tdTomato^+^ (17.79%) cells was confirmed by immunohistochemistry and flow cytometry just in 1/6 tumors in animals injected with tdTomato^−^ non-CSC cells (Figure 4).

### 2.2. Involvement of AKT2 in Dynamism of EMT and CSC-Like Phenotype

The observation of relatively high incidence of tumors derived from MDA-MB-231 tdTomato^−^ cells, with low incidence of “new” tdTomato^+^ CSC-like cells, made us wonder whether there is another type of phenotypic dynamism, for instance EMT, involved in the process of tumorigenesis. Notably, we observed more EMT genes (low *E-CADHERIN*, high *FIBRONECTIN*, and high *VIMENTIN*) in the tumors derived from non-CSC compared to tumors derived from CSC-like tdTomato^+^ cells (Figure 5A). *NOTCH4*, an important EMT and CSC marker, was also highly over-expressed in the tumors derived from MDA-MB-231 non-CSC. Besides, *mTOR*, an *AKT2* downstream gene was also over-expressed in these tumors, compared to tumors derived from tdTomato^+^ cells (Figure 5A).

TWIST, an AKT downstream gene, has been previously described as responsible for mesenchymal to epithelia switch. Even though injected MDA-MB-231 cells completely lack *TWIST* expression, 5/6 (83%) tumors derived from tdTomato^−^ MDA-MB-231 cells expressed *TWIST* (Figure 4B). In the group of CSC-like injected animals, just 3/7 (42%) express *TWIST* in the resultant tumors (Figure 4, Figure 5B). *AKT2* mRNA expression level was however not changed within the different groups (Appendix A). Of note, we detected a significant increase of *AKT2* mRNA expression (FC = 2.3 ± 0.32, *p* = 0.01) in circulating tumor cells (CTC) isolated from the blood of tumor bearing animals. In addition, CTCs derived from tumors also expressed high levels of *mTOR* (FC = 2.3 ± 0.26, *p* = 0.02) and *TWIST* (FC = 2.6 ± 0.28, *p* = 0.03) compared to primary tumors (Figure 5C). This was accompanied by significant decrease of *E-CADHERIN* expression (FC = 0.07 ± 0.13, *p* = 0.02) (Figure 5C).

Further exploration of the AKT2/TWIST/mTOR axis in mammospheres, resembling CTCs in vitro, revealed MDA-MB-231 cells in low attachment expressed significantly higher level of *AKT2* (FC = 2.4 ± 0.19, *p* = 0.00003) and *mTOR* (FC = 1.9 ± 0.05, *p* = 0.0003), compared with the same cells cultured in monolayer (Figure 5D). Moreover, these gene expression levels decreased when the cells were cultured again in attachment conditions (Figure 5D). Expression of *TWIST* mRNA was not detected in these conditions.

Interestingly, *AKT2* expression was not altered in CSC fraction either in MDA-MB-231 cells or in MCF7 cells. In fact, none of the *AKT* isoforms (*AKT1*, *AKT2*, and *AKT3*) were differentially expressed in the CSC fraction compared with the non-CSC fraction (Appendix A).

### 2.3. Consequences of AKT2 Inhibition for CSC-Like/EMT Phenotype

The increased expression of EMT genes (*AKT2*, *TWIST*, and *mTOR*) in tdTomato^−^ tumors, in CTCs and in mammospheres, highlights the importance of AKT2 in tumor growth and CSC malignancy. To pursue this line of investigation further, we explored the consequences of *AKT2* silencing in three different cancer cell lines, and their CSC-like fractions (MDA-MB-231, HCT8, and MCF7). Even though the suitability of AKT2 targeting in general terms is well known, its role in CSC malignancy has not been elucidated yet [17,18,19,25,32].

Since AKT2 is known to play an important role in EMT, and is a likely candidate for driving the CSC phenotype, we first characterized mesenchymal gene expression levels in the investigated cells lines. CSC isolated from the mesenchymal cell lines MDA-MB-231 did not differ in EMT gene expression that was high in both “bulk” fraction as well as in CSC-like fraction (Figure 6A). In contrast, CSC isolated from epithelial cell lines (MCF7, HCT8) showed increased mesenchymal phenotype (low *E-CADHERIN*, high *VIMENTIN*, and *FIBRONECTIN*) compared to their respective “bulk” counterparts (Figure 6A). In addition, mesenchymal, stem-like cell lines SKBR3 and MDA-MB-468 were included in the analysis, as well.

Specific inhibition of *AKT2* by siRNA reduced *AKT2* mRNA expression by 80% in all cell lines used, while no effect on *AKT1* isoform expression was detected (Appendix A). Downregulation of AKT2 was also confirmed at the protein level by Western Blot (WB) (Figure 7).

As expected, sorted CSC subpopulations showed significantly higher capacity for anchorage-independent growth compared to non-CSC fractions, further confirming the CSC nature of tdTomato^+^ cells (Figure 6B). Similarly, CSC from MDA-MB-231 and MCF7 showed significantly higher invasive potential. No difference in invasion was detected between CSC and non-CSC in HCT8 cell line (Figure 6C).

For MDA-MB-231 and HCT8, *AKT2* silencing significantly reduced the number of cells able to grow without anchorage, and the invasive potential of both CSC and non-CSC. In the case of MCF7, anchorage-independent growth capacity and invasive ability was strongly reduced after *AKT2* silencing only in the CSC subpopulation (Figure 6B,C). Moreover, anchorage-independent growth and invasion were significantly reduced also in breast cancer cell lines with clear mesenchymal phenotype; with high expression of ALDH1A1, (MDA-MB-468 and SKBR3) after *AKT2* silencing (Appendix A). Furthermore, we observed a significant impairment of cell migration and proliferation of the mesenchymal cells in which *AKT2* was silenced, in MDA-MB-231, MDA-MB-468, and SKBR3 cell lines (Appendix A). In contrast, no difference in the migration and proliferation of “bulk” epithelial MCF7 cells was observed after *AKT2* silencing (Appendix A).

To explain mechanistically the downstream effect of *AKT2* silencing in different cell types, and in CSC-like cells derived from these cell lines, is however not straightforward. We observed different pathway involvement depending on the cellular background. In mesenchymal (MDA-MB-231) as well as epithelial (HCT8) cell lines that do not express *TWIST* in vitro, the predominant effect on *AKT2* inhibition was de-regulation of mTOR signaling (*p.53*, *GSK3b* high) (Figure 7A), whereas in mesenchymal cell lines that do express *TWIST* (MDA-MB-468, SKBR3), and in mesenchymal CSC isolated from epithelial MCF7 cells, *AKT2* inhibition caused the reversion of mesenchymal to epithelial phenotype (*E-CADHERIN* high, and *N-CADHERIN*, *VIMENTIN*, and *FIBRONECTIN LOW*; Figure 7B,C). No clear effects on EMT genes and mTOR pathway signaling were observed in MCF7 bulk cells, even though *TWIST* and *mTOR* mRNA were downregulated in these cells after *AKT2* inhibition (Appendix A).

Furthermore, we observed also down-regulation of *E-CADHERIN* and some stem cell genes (*OCT4, NANOG*) after *AKT2* silencing in MDA-MB-231 cells, even though the expression of *N-CADHERIN*, *VIMENTIN*, and *FIBRONECTIN* was not changed in these mesenchymal cell lines, conforming tide connection between EMT and CSC phenotype (Appendix A).

Finally, we tested the capability of an AKT2-specific inhibitor, CCT128930, to inhibit mammosphere growth, as a proof of concept for the further development of nano-therapies based on AKT2 inhibition and the resulting EMT and CSC phenotype, blockade. Furthermore, *AKT2* inhibition using siRNA was not optimal in low attachment cell culture, but CCT128930 be used in these conditions. However, when inhibiting AKT2 with CCT128930, we observed a progressive cell death in a dose dependent manner, that was quantified using DAPI (4′,6-Diamidine-2′-phenylindole dihydrochloride) vital staining (Figure 8).

## 3. Discussions

The MDA-MB-231 cell line is a model of highly aggressive TNBC. This cell line is categorized as a mesenchymal stem cell-like cell line and displays high expression of CD44 and low expression of CD24, which is a common signature of breast CSC and at the same time it is an indication of the mesenchymal origin of these cells. Besides, MDA-MB-231 cells are highly proliferative with elevated migratory capacity, and as few as 1000 cells can produce a tumor in vivo with metastatic dissemination. To our best knowledge, the MDA-MB-231 CSC model has not been described previously either in vitro or in vivo. This is probably due to the high complexity of this cell line and no clear boundaries between the EMT and CSC phenotype of MDA-MB-231 cells. CSC within this cell line had been identified just by surface markers, specific inhibition, or their adhesion properties [33,34,35,36]. We were however able to distinguish a unique population of CSC-like cells within the bulk cell line based on ALDH1A1-tdTomato expression vector. Accordingly, tdTomato^+^ CSC-like cells express stem cell markers and have higher invasive and transformative potential compared to tdTomato^−^ non-CSC [7,33].

Nevertheless, we detected a high level of phenotype dynamism in the tdTomato^−^ cells, corresponding to the non-CSC fraction. These cells were also able to form mammospheres in low attachment conditions, even though in a lower extent than tdTomato^+^ CSC-like cells. Interestingly, the non-CSC fraction became tdTomato^+^ during the process and CSC genes expression was significantly increased, confirming once again the close EMT–CSC connection. The phenomenon of dynamic CSC phenotype was initially predicted by a mathematical Markov model and was later corroborated experimentally in vitro and in vivo [3,5,37,38,39]. Accordingly, after successful anti-CSC treatment, the remaining non-CSC tumor cells can change their phenotype to CSC-like phenotype in order to ensure tumor progression [5,6,7]. Similar behavior was observed also in vivo; tdTomato^−^ cells originated a tumor containing tdTomato^+^ cells. However, this occurred just in one out of six in vivo models. In the rest of the tumors we did not detect clear tdTomato expression, although these tumors showed increased EMT gene expression and *NOTCH4* expression. Evidently, there was an adaptation of the phenotype; since we detected “de novo” *TWIST* expression despite the fact that MDA-MB-231 cells before injection did not express this gene. Since other EMT genes (low E-*CADHERIN*, high *VIMENTIN*, and N-*CADHERIN*) and *mTOR* were also overexpressed in these tumors, we speculated whether AKT2, the mastermind of TWIST and mTOR was responsible for these phenotypic changes. *AKT2* and *mTOR* was also overexpressed in mammosphere cell culture, however, their expression as well as stem cell marker expression, lowered after returning the cells to attachment conditions. This observation supports dynamism based on growth conditions, in order to survive [5,6,7]. AKT2 is a well-known target for cancer treatment, and we have demonstrated its importance for CSC growth and dynamism in breast cell lines, suggesting that AKT2 could be an ideal candidate for an effective anti-cancer therapy [12,30]. We have provided evidence that the inhibition of *AKT2* not only affects “bulk” tumor cells, but also inhibits CSC survival in low attachment conditions. Additionally, *AKT2* down-regulation inhibited the invasion and transformation capacity of both cells (bulk and CSC-like cells). Since invasiveness is related to several distinct cellular functions including adhesion, motility, and detachment, the regulation of EMT should hamper the invasive properties of cells [40]. Malignant transformation occurs via a series of genetic and epigenetic alterations that allow the tumor cell population to proliferate independently of both external and internal signals that normally restrain growth. Likewise, CSCshow reduced requirements for adherent-dependent growth and are not restricted by cell–cell contact [41]. The increased expression of E-CADHERIN caused by EMT reversion, overturns this capacity for unregulated proliferation. Accordingly, invasiveness and malignant transformation of cells was affected by *AKT2* silencing only in mesenchymal cells (CSC, bulk cells), while no changes were observed in the epithelial MCF7 cell line. Similarly, proliferation and migration of “bulk” epithelial MCF7 was not affected, while mesenchymal cells (MDA-MB-231, MDA-MB-468, and SKBR3) showed decreased cell migration and proliferation after *AKT2* inhibition. Since MCF7 cells normally display an epithelial phenotype (high *E-CADHERIN*, low *FIBRONECTIN*, and low *VIMENTIN*), MET (mesenchymal-to-epithelial transition) triggered via *AKT2* silencing had no effect on “bulk” MCF7 cell population. However, CSCsubpopulations isolated from MCF7 epithelial cell line presented a more mesenchymal phenotype than the non-CSC, mainly by a decrease in *E-CADHERIN* expression, and an increase in *VIMENTIN* and *FIBRONECTIN* expression. In this mesenchymal CSC fraction (1–2%) isolated from epithelial MCF7 cells we witnessed the impact of *AKT2* silencing in terms of reduced transformation and invasive capacity. Since these CSC are thought to be responsible for tumor growth and maintenance, curtailment of their tumorigenic capacity could prevent tumor progression. We speculated that an effective therapy could consist of standard therapy in combination with *AKT2* inhibition, to completely eradicate all cancer cells.

Furthermore, a high level of *AKT2* expression was observed also in circulating tumor cells (CTCs) in vivo. Thus, we hypothesized that targeting AKT2 could be effective in multiple levels; it may diminish invasiveness and transformative potential of CSC, to revert the mesenchymal phenotype of aggressive tumor cells back to a more epithelial type, and to reduce of the incidence of CTCs in the bloodstream and the concomitant opportunity for metastatic spread.

In this study we focused on AKT2, since it is the only isoform of AKT associated with EMT and tumor dissemination. Besides, its over-expression positively correlates with poor survival rates. The role of AKT1 and AKT3 in breast cancer should be studied as well, because these isoforms have distinctive and non-redundant activities [15,21,42]. In particular, expression of *AKT3* has been recently associated with TNBC tumors [22]. Yet, its role in tumor progression is controversial as some studies have showed an inhibitory effect of AKT3 on cell migration and invasion [19].

Our data did not allow us to reach a clear conclusion regarding the downstream pathways that were affected in these processes by targeting *AKT2*. Since the AKT2/TWIST/mTOR signaling axis was involved in many processes, its exact mode of action depended on the cellular background, the clonal evolution, and CSC state among others. It is likely that *AKT2* inhibition could affect a broad spectrum of downstream pathways; although there are always some common cellular responses: (i) An inhibition of growth, (ii) an inhibition of invasiveness, and (iii) an inhibition transformation potential. Therefore, inhibition of AKT2 could potentially be a good candidate for breast cancer treatment and should be further analyzed to probe its therapeutic use. In fact, numbers of small molecule inhibitors targeting various components of the PI3K/AKT pathway are already in clinical development and evaluation (e.g., Afuresertib (GSK2110183), AZD5363, BAY1125976, Ipatasertib (GDC-0068), MK-2206, etc.). Given the multiple feedback loops within the PI3K-AKT signaling pathway, dual inhibition at different levels is also under investigation (e.g., gefitinib, erlotinib, and ridaforolimus with MK-22069, respectively). Despite the demonstrated clinical benefits of these treatments, their over-lapping toxicities preclude required dosage. One method to overcome current drawbacks in regards of toxicity and effective dosage is the development of delivery systems for siAKT2, and AKT2 inhibitors targeted to malignant cells [43,44]. Nanomedicine can facilitate the administration of hydrophobic agents and/or siRNA, while protecting them during circulation, lowering the undesired side effects associated with systemic, non-controlled distribution of the drug. Nanomedicines can also evade multi drug resistant mechanisms (i.e., MDR channels) and increase drug’s intracellular accumulation, improving the efficacy of the treatment [45,46]. More advanced delivery systems could be used to combine several inhibitors of AKT2/TWIST/mTOR pathway to synergize treatment efficacy, improve clinical outcomes, and overcome resistance to treatments, while protecting somatic cells from undesired toxicities.

## 4. Materials and Methods

### 4.1. Cell Culture

Breast (MCF-7, MDA-MB-231, MDA-MB-468, and SKBR3) and colon (HCT8 and HCT116) cancer cell lines were obtained from American Type Culture Collection (ATTC, LGC Standards, Barcelona, Spain; Appendix A). Detailed culture conditions are described in the Appendix A.

### 4.2. Generation of CSC Models

CSC models were generated from breast and colon tumor cell lines as previously reported by our group [7]. Briefly, tdTomato reporter cDNA was cloned under the minimal ALDH1A1 promoter using a pENTRtm 5′-TOPO^®^ TA Cloning Kit (Thermo Fisher Scientific, CA, USA) and tdTomato fluorochrome was used as a reporter of ALDH1A1 promoter activity. Cells were transfected with Lipofectamine 2000 (Thermo Fisher Scientific, CA, USA), and cultured under selective pressure with 10 μg/mL Blasticidin for two weeks. Positive tdTomato cells (tdTomato^+^) were sorted by fluorescence-activated cell sorting (FACS) and reseeded to reproduce the parent cell line in which tumoral non-CSC showed no expression of tdTomato (tdTomato^−^). Expression of known stemness markers (*ALDH1A1, NOTCH4*, *POU5F1/OCT4*, *ALOX5*, *CMKLRI*, *ABCG2*, *CXCR1*, *CD44*, and *CD24*) was used to determine the stemness profile of cells.

Further, tdTomato^+^ cells were tested for their capacity to form tumorspheres when growing in non-attachment conditions. For this, a maximum of 10.000 cells/mL were cultured 7–10 days in ultra-low attachment plates in serum-free RPMI 1640 Medium (1X, Life Technologies) supplemented with 6% glucose, 10 uL/mL L-GglutamaxTM, 10 uL/mL antibiotic-antimitotic mixture (Thermo Fisher Scientific, CA, USA), 4 µg/mL heparin (Sigma, MI, USA), 2 mg/mL BSA, 0.02 ug/mL EGF (Sigma, MI, USA), and 0.01 µg/mL FGFb (Thermo Fisher Scientific, CA, USA), and 10 µg/mL putrescin, 0.1 mg/mL apo-transferrin, 25 µg/mL insulin, 30 µM selen, and 20 µM progesterone (all from Sigma, MI, USA).

We used p.96 low attachment plates to grow the exact number of cells (1, 3, 10, 30, 100, 300, and 1000) sorted by FACS.

Similarly, sorted tdTomato negative (tdTomato^−^) populations were cultured in non-attachment conditions for 17 days, and plated again into normal culture dishes for a further four days (diagrammatic representation in Figure 4A).

### 4.3. Fluorescence-Activated Cell Sorting (FACS)

FACS was used to sort CSC and non-CSC subpopulations from a heterogeneous population of MCF-7, MDA-MB-231, and HCT-8 cells. For cell sorting, a starting amount of 5X106 cells were used. Cells were detached and resuspended in phosphate buffered saline (PBS; Lonza, Porriño, Spain), supplemented with 10% Fetal Bovine Serum (FBS). DAPI (1 μg/mL, Life Technologies) was used for vital staining (Life Technologies). Cells were sorted according to tdTomato and DAPI expression in a FACS Aria cell sorter (BD Biosciences, CA, USA). Sorted cells were collected in complete medium without antibiotic and used for subsequent experiments.

### 4.4. RNA Extraction and Quantitative RT-PCR (qRT-PCR)

Total RNA was extracted from cells using an RNeasy Micro Kit (Qiagen, BCN, Spain) and the RNA obtained was reverse transcribed using a High Capacity cDNA Reverse Transcription Kit (Thermo Fisher Scientific, CA, USA) according to the manufacturer′s instructions. The cDNA reverse transcription product was amplified with specific primers (Appendix A) by qPCR using a SYBR Green method (Thermo Fisher Scientific, CA, USA). The reaction was performed in triplicate on a 7500 Real time PCR system (Thermo Fisher Scientific, CA, USA). Actin and S18 were used as endogenous controls. Relative mRNA levels were calculated using the comparative Ct method (2e^−ΔΔCt^) [47].

### 4.5. In Vivo Tumorigenic and Metastatic Capacity Assay

Female NOD.CB17-Prkdcscid/J mice (Charles River, MA, USA) were kept in pathogen-free conditions and used at six weeks of age. Animal care was handled in accordance with the Guide for the Care and Use of Laboratory Animals of the Vall Hebron University Hospital Animal Facility, and the experimental procedures were approved by the Animal Experimentation Ethical Committee at the institution (approval number CEA-OH/9467/2). All the in vivo studies were performed by the Unique Scientific and Technical Infrastructures (ICTS) “NANBIOSIS”, more specifically at the Bioengineering, Biomaterials and Nanomedicine Research Center (CIBER-BBN) in vivo Experimental Platform of the Functional Validation and Preclinical Research (FVPR) area (http://www.nanbiosis.es/portfolio/u20-in-vivo-experimental-platform/; Barcelona, Spain). MDA-MB-231.Fluc2-C19.ALDH1-tdTomato^+^ at 1000 cells in 50 µL sterile PBS: Matrigel (1:1) were inoculated orthotopically into the right mammary fad pad (IMFP; *n* = 8). Tumor growth was monitored twice a week by conventional caliper measurements (D × d2/2, where D is the major diameter and d the minor diameter). Once primary tumors reached a tumor volume range between 250–450 mm^3^, tumors were excised, weighted, and divided into several fragments for FACS analyses. One additional sample was fixed with 4% formaldehyde and paraffin embedded for histopatological analysis. At termination, blood samples were drawn from each animal by cardiac puncture. Whole blood was transferred directly into commercial EDTA containing tubes and processed immediately to isolate circulating tumor cells (CTC). Briefly, blood samples were subjected to several cycles of hemolysis using mixture of 90% of 16 M NH4Cl and 10% of 0.17 M Tris (pH 7.65) and centrifuged. White blood cell pellets were then examined by flow cytometry (Fortessa, BD Biosciences, CA, USA). Data were analyzed with FCS Express 4 Flow Research Edition software (De Novo Software, Glendale, CA, USA).

### 4.6. AKT2 Inhibition

siRNAs were designed by Shanghai Gene Pharma (Shanghai, China). The sense AKT2 siRNA sequence used was 5′-GCUCCUUCAUUGGGUACAATT-3′, while a non-specific sequence 5′-UUCUCCGAACGUGUCACGUTT-3′ was used as a control (control siRNA). The siRNAs were transfected into cells using Lipofectamine 2000 (Thermo Fisher Scientific, CA, USA). Transfections were done in six-well plates with 2 × 10^5^ cells per well, with a final concentration of 50 nM of siRNA in antibiotic-free medium. The medium was changed 6 h after the transfection and cells were harvested 72 h after transfection. In the case of the non-CSC to CSC reversion experiments, AKT2 was inhibited with the specific inhibitor CCT128930. Briefly, sorted MDA-MB-231 tdTomato^−^ cells were plated in serum-free low attachment conditions, in the presence of 6.25 µM, 12.5 µM and 25 µM of CCT128930 (Selleckchem, TX, USA) and the formation of tumorspheres was monitored over time.

### 4.7. Protein Extraction and Western Blotting

Cell pellets were lysed with Cell Lytic M reagent (Sigma, MI, USA)) containing a protease inhibitor cocktail (Sigma, MI, USA). Proteins in crude lysates were quantified using the bicinchoninic acid assay (BCA) Protein Assay (Pierce Biotechnology, MA, USA)). A total of 20 μg of whole-cell lysates were separated by SDS-PAGE and transferred onto nitrocellulose membranes (Merck Millipore, Madrid, Spain). Blots were probed using primary antibodies (Appendix A). Proteins were detected using horseradish peroxidase (HRP)-conjugated secondary antibodies, anti-mouse (P0447, Dako, Barcelona, Spain) or anti-rabbit (P0217, Dako, Barcelona, Spain), incubated for 1 h at room temperature. Band intensity on the blots was quantified using the GeneTools Program (SynGene, Cambridge, UK).

### 4.8. Cell Transformation Assay (Anchorage-Independent Growth Assay)

Anchorage-independent growth of the different cancer cell lines was assessed using the CytoSelect™ Cell Transformation Assay Kit (Cell Biolabs, CA, USA). A semi-solid agar media was prepared accordingly to the manufacturer’s instructions. Lipofectamine-AKT2-siRNA or Lipofectamine-Control-siRNA was added to each well respectively. After 6–8 days of incubation the colonies were observed under the microscope and viable transformed cells were counted using trypan blue.

### 4.9. Invasion Assay

The invasiveness of MDA-MB-468 and SKBR3 cells was assessed using the CytoSelect™ Laminin Cell Invasion Assay Kit (Cell Biolabs, CA, USA). The polycarbonate membrane inserts were coated with a uniform layer of Laminin-I and used to discriminate invasive from non-invasive cells. Whereas invasive cells are able to degrade the laminin matrix layer and pass through the pores of the polycarbonate membrane to the bottom of the insert membrane, non-invasive cells remain in the upper chamber. Briefly, inserts were placed in a 24 well plate and cell suspensions containing 1 × 10^6^ cells/mL previously transfected with AKT2 siRNA or control siRNA, added to the insert. After 48 h incubation, invasive cells were dissociated from the lower side of the membranes, lysed, and quantified using CyQuant^®^ GR Fluorescent Dye (Cell Biolabs, CA, USA).

#### Statistical Analysis

Statistical analysis was performed using unpaired Student′s *t*-test. Differences were regarded as statistically significant when the *p*-value was smaller than 0.05.

## 5. Conclusions

Advanced cancers adapted very well to therapeutic challenges in order to survive under stress conditions. Cells within the tumor often underwent dynamic phenotypic changes. We provided here a CSC fluorescent model of the MDA-MB-231 cell line, in which these constant changes could be observed in vitro as well in vivo. We focused our attention on AKT2 expression, which was elevated in low attachment growth conditions of mammospheres and in circulating tumor cells. We suggest that AKT/TWIST/mTOR pathway should be targeted to affect (i) CSC survival, (ii) “bulk” tumor cells, and (iii) the non-CSC to CSC reversion phenotype (using additional CSC cell lines).

## Figures and Tables

**Figure 1 cancers-11-01058-f001:**
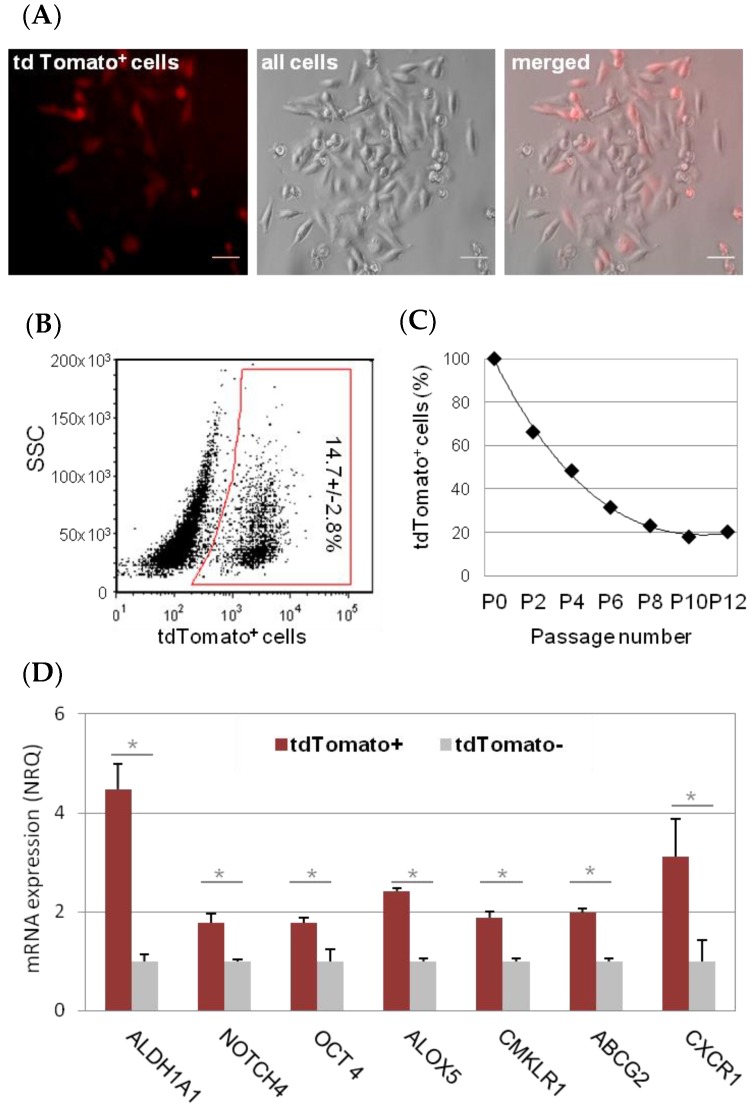
MDA-MB-231-ALDH1A1/tdTomato cancer stem cell (CSC) model. After transfection with reporter vector ALDH1A1/tdTomato, CSC-like cells express fluorescent reporter (tdTomato) under the CSC-specific promoter (ALDH1A1). Scale bar represents 20 µm (**A**). This allows the CSC quantification and sorting by fluorescence-activated cell sorting (FACS) (**B**). Sorted tdTomato^+^ cell population dropped and stabilized over passages (**C**). We confirmed by qPCR, that tdTomato^+^ cells (CSC) express stem cell markers. Results are expressed as NRQ (relative normalized quantities) mean ± SEM (*n* ≥ 3); * *p* < 0.05; ** *p* < 0.01, *** *p* < 0.001 (**D**).

**Figure 2 cancers-11-01058-f002:**
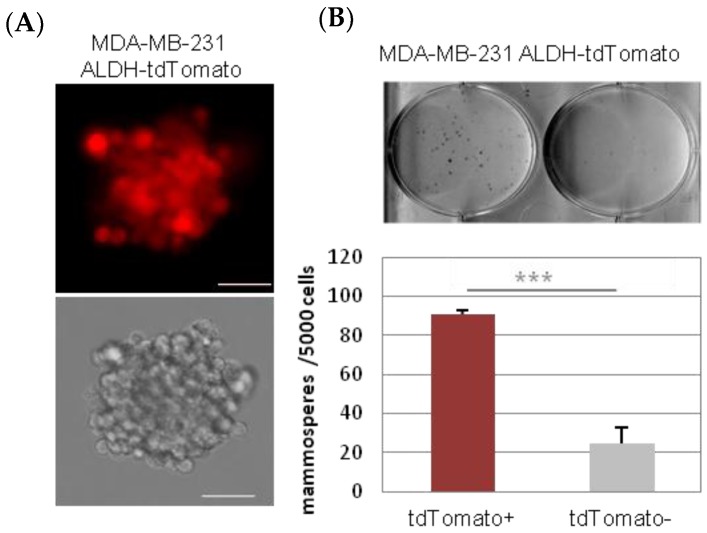
Mammosphere culture of tdTomato^+^ cells. tdTomato^+^ CSC were able to form mammospheres in low attachment plates, in serum free media. Scale bar represents 100 µm (**A**) tdTomato+ cells formed significantly more mammospheres than tdTomato^−^ cells (**B**). *** *p* < 0.001

**Figure 3 cancers-11-01058-f003:**
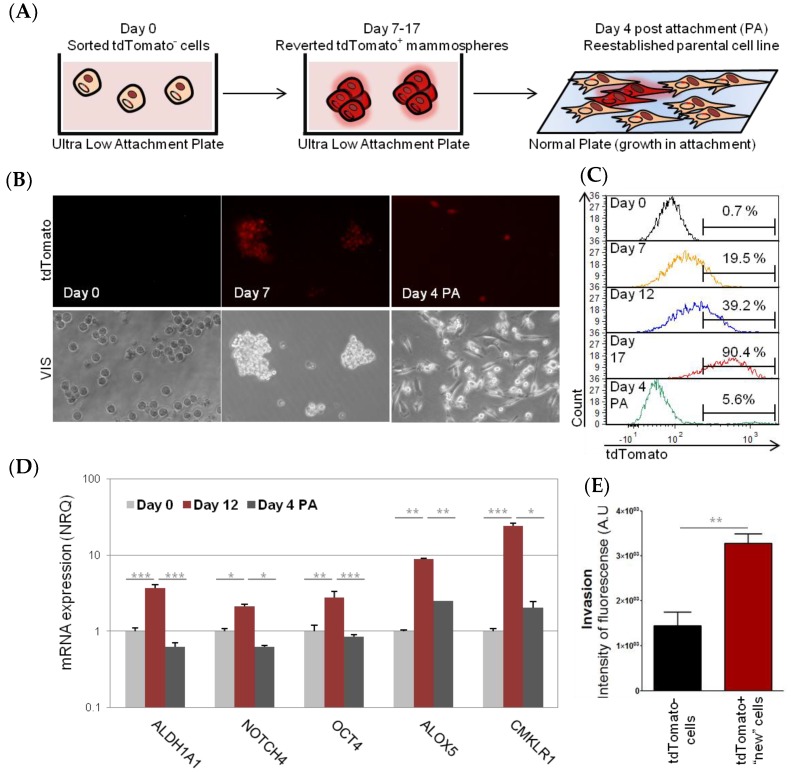
MDA-MB-231 reversion process. Schematic representation of the experiment. Briefly, tdTomato^−^ (non-CSC) were placed in low attachment plates in FBS (Fetal Bovine Serum)-free culture medium and were cultured for 7–17 days. During this time cells acquired the CSC phenotype. Afterwards, tdTomato^+^ cells were plated back into adherent plates in complete culture medium and were allowed to grow for another four days (**A**). The cells were monitored over the time by fluorescent microscopy (**B**) and flow cytometry (**C**). The stem cell properties of “new” tdTomato^+^ cells were confirmed by a change in stem cell gene expression profile, measured by qPCR (**D**) and by an increase of invasive ability (**E**). Results are expressed as NRQ (relative normalized quantities) mean ± SEM (*n* ≥ 3); (* *p* < 0.05; ** *p* < 0.01, *** *p* < 0.001).

**Figure 4 cancers-11-01058-f004:**
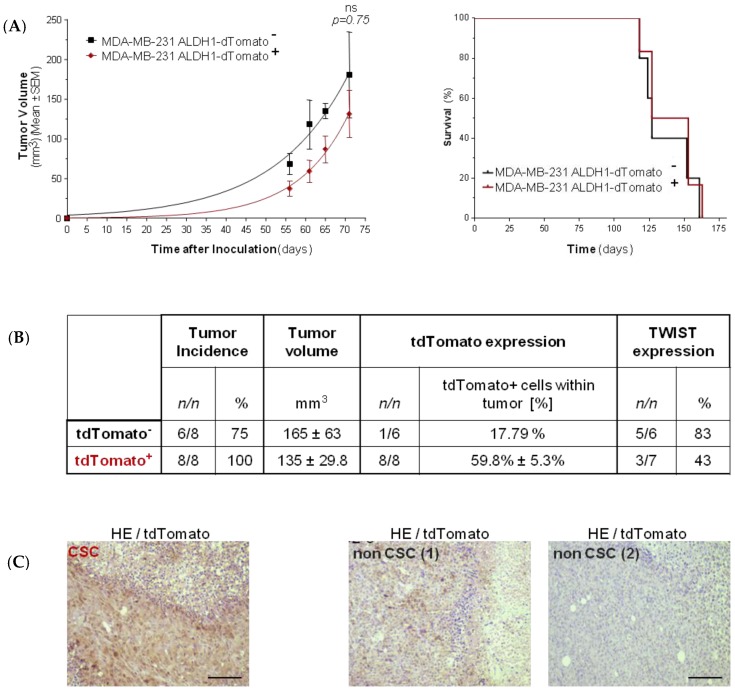
MDA-MB-231 in vivo model. Tumor growth and Kaplan–Meier survival curves of mice inoculated inoculated orthotopically into the right mammary fad pad (IMFP) with MDA-MB-231dTomato^−^ non-CSC (*n* = 6), and MDA-MB-231dTomato^+^ CSC (*n* = 8) cell variants was monitored over time. No statistically significant differences were detected (**A**). Summary of information regarding tumor incidence, tumor growth, and tdTomato and TWIST expression within the tumors (**B**). Examples of IHC (immunohistochemistry) (HE/td Tomato) in representative tumors of animals inoculated with CSC-like cells, and non-CSC (tdTomato positive as well as the tdTomato negative example). Magnification bar represents 100 μm (**C**).

**Figure 5 cancers-11-01058-f005:**
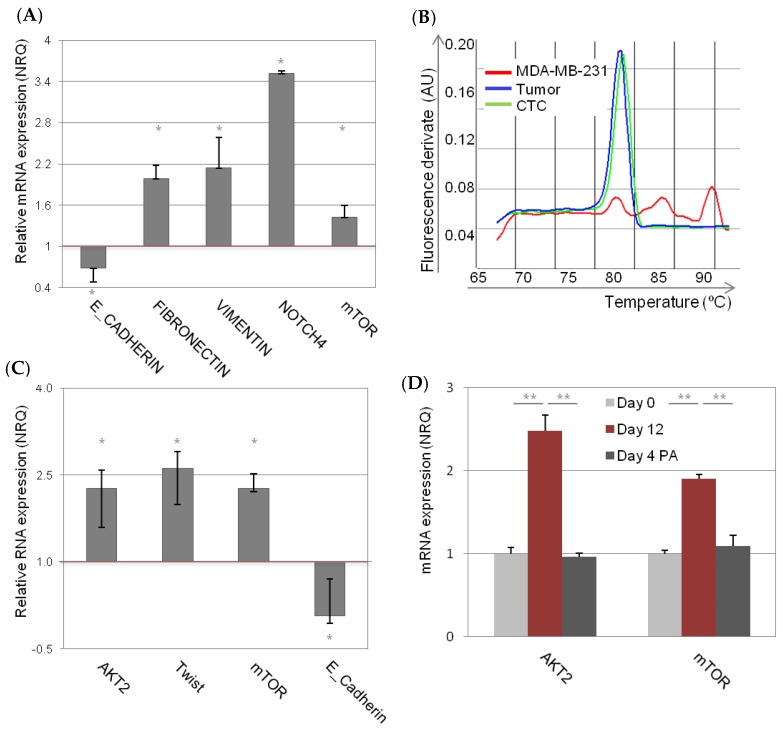
AKT2/TWIST/mTOR axis in vivo and in vitro. Gene expression in the tumors derived from non-CSC relative to tumors derived from CSC-like tdTomato^+^ cells (**A**). Dissociation curve plot confirmed that tumors expressed *TWIST*, even though the injected cells did not express *TWIST* per se (**B**). Gene expression in circulating tumor cells (CTCs) isolated from the blood of tumor bearing animals relative to gene expression of tumors (**C**). Gene expression in low attachment culture conditions, relative to the same cells cultured in monolayer (**D**). qPCR results are expressed as NRQ (relative normalized quantities) mean ± SEM (*n* ≥ 3); (* *p* < 0.05; ** *p* < 0.01).

**Figure 6 cancers-11-01058-f006:**
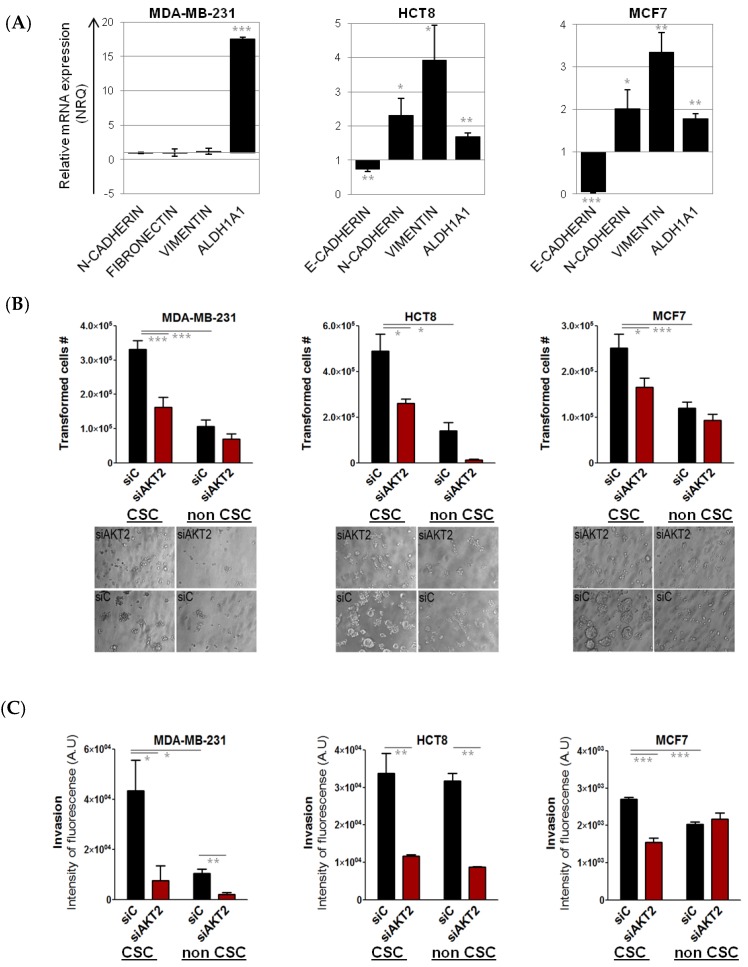
Effects of AKT2 silencing in MDA-MB-231, HCT-8, and MCF-7 CSC. MDA-MB-231, HCT-8, and MCF-7 CSC were separated from non-CSCs and the relative EMT gene expression profile was quantified by qPCR. The presented values are relative to their non-CSC population (**A**). Anchorage-independent growth assay: The graph represents the number of cells capable of anchorage-independent growth, comparing CSCs and non-CSC transfected with siAKT2 and control siRNA (siC). Images are photographs of colonies formed after seven days of incubation with siAKT2 and siC (**B**). Invasion assay: The graph represents the number of cells that demonstrated invasive capacity comparing CSC and non-CSC transfected with siAKT2 and siC (**C**). Results are expressed as NRQ (relative normalized quantities) mean ± SEM (*n* ≥ 3); (* *p* < 0.05; ** *p* < 0.01, *** *p* < 0.001).

**Figure 7 cancers-11-01058-f007:**
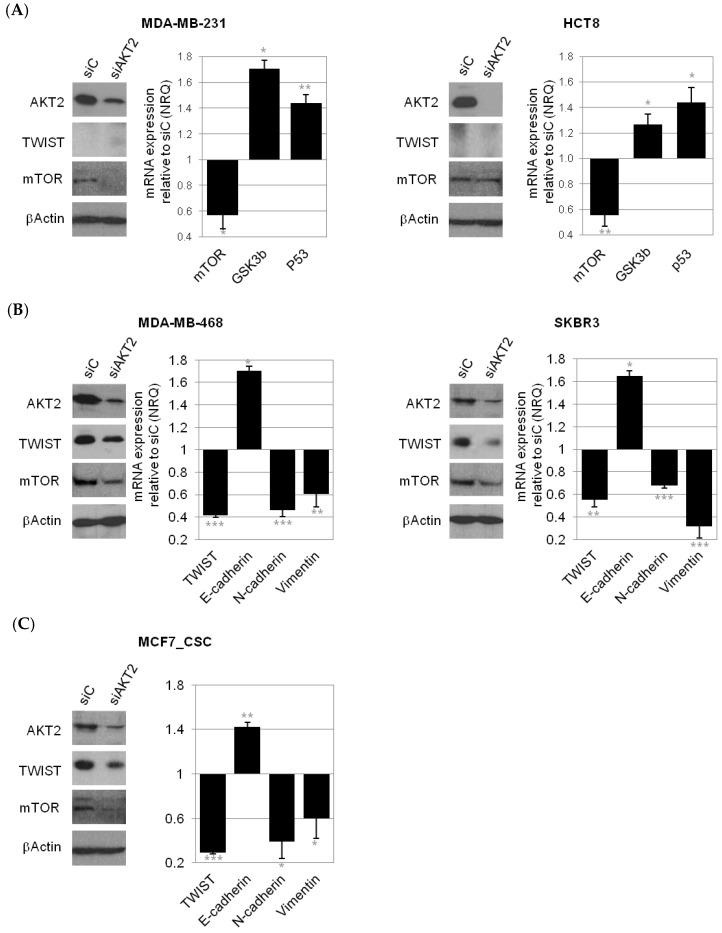
AKT2 silencing on gene expression. qPCR quantification of gene expression levels of key regulators in the mTOR signaling pathway in cells that do not express TWIST (MDA-MB-231, HCT8; (**A**) and epithelial-to-mesenchymal transition (EMT) signaling pathway in mesenchymal cells that do express TWIST (MDA-MB-468, SKBR3; (**B**) and (MCF-7 CSC) after the treatment with siAKT2 (**C**). The presented values are relative to cells transfected with siC and are expressed as NRQ (relative normalized quantities) mean ± SEM (*n* ≥ 3); (* *p* < 0.05; ** *p* < 0.01, *** *p* < 0.001).

**Figure 8 cancers-11-01058-f008:**
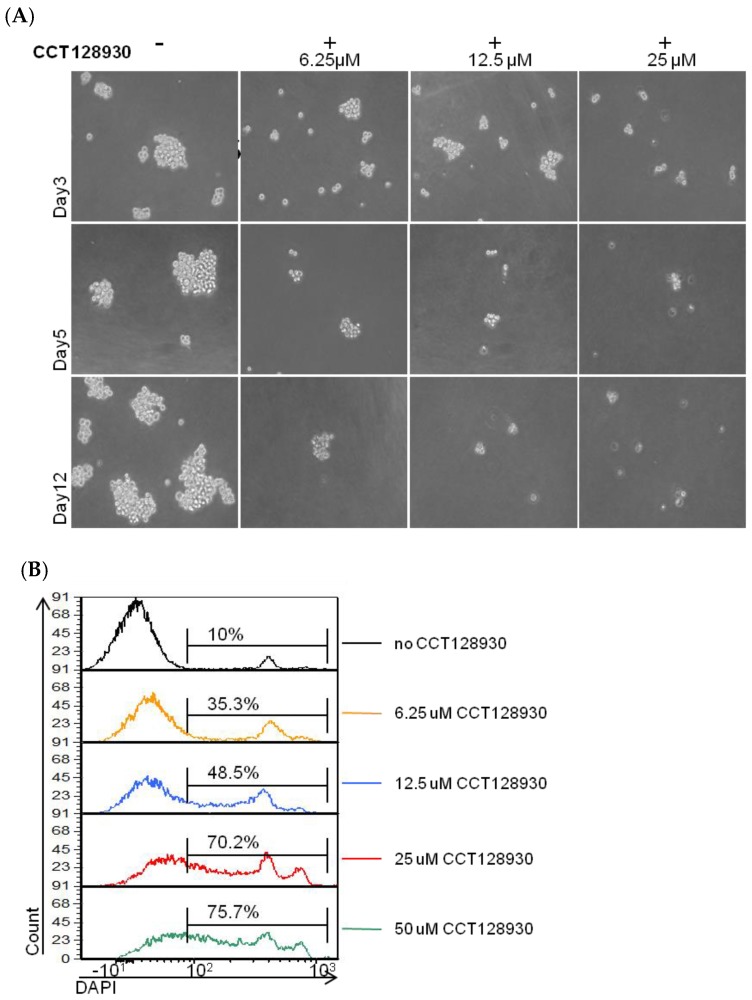
Effect of AKT2 chemical inhibition (CCT128930) on mammosphere growth. Sorted MDA-MB-231 tdTomato− cells were plated in serum free low attachment conditions, in the presence of 6.25 µM, 12.5 µM, and 25 µM of CCT128930 and the formation of tumorspheres was monitored over time. With increasing concentrations of CCT128930, there were fewer tumorspheres, of smaller size, (**A**) and we detected increasing proportions of dead (DAPI+) cells by flow cytometry (**B**).

**Table 1 cancers-11-01058-t001:** Incidence of mammosphere formation.

Initial Number of Cells Per Well	Stemness Type	1	3	10	30	100	300	1000
Mammosphere formation	CSC-like	−	−	+	+	+	+	+
	Non-CSC	−	−	−	−	+	+	+
Incidence of mammosphere formation (*n*/*n*)	CSC-like	N/A	N/A	6/12	8/12	12/12	12/12	12/12
	Non-CSC	N/A	N/A	N/A	N/A	1/8	4/10	7/12
Incidence of mammosphere formation (%)	CSC-like	N/A	N/A	50%	67%	100%	100%	100%
	Non-CSC	N/A	N/A	N/A	N/A	12%	50%	58%

*n*/*n*—number of wells with mammospheres *per* number of wells in total; N/A—not applicable.

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
