# Peer review of "Pivotal Role of AKT2 during Dynamic Phenotypic Change of Breast Cancer Stem Cells"

_cancers, 2019, doi:10.3390/cancers11081058_

Round 1

Reviewer 1 Report

Authors attempted to develop an innovative regimen for treating breast cancer, particularly Triple Negative subtype.  Despite the role that Akt-2 plays on breast neoplasm and on colon cancers have attracted ample research attention, the clinical application remains somewhat scant as of today.

While the experimental approaches and the evolving  data were scientific sound, the following attention shall be brought to authors’ attention.

In light of the finding in which Si-AKT2 resulted in hampered invasive ability as well as lowered anchorage-independent growth, shall the similar approach be eventually expanded to evaluate its effects on tumor formation in laboratory animals as Table 2 did?

Upon suppressing AKT2 activity, which of the two, Si-AKT2 or inhibitory agent CCT128930, exerts a far better treatment benefit, clinically in animal models?  The Western blot showing diminished AKT2 from the CCT128930 shall be presented in the result section.

Likewise, will CCT128930 render non-specific inhibition or side effects on genes other than AKT-2?  Where was CCT128930 obtained from has not been described in the manuscript.

Typically, monotherapy works not as effective as combination treatments. Shall authors speculate if combining with inhibitors for mTOR  (using Rapamycin) or for TWIST could synergize treatment outcomes?

At the conclusion of the article, the specific subtype of breast cancer that are expected to benefit from AKT-2 inhibition remains unclear.   Even though triple-negative MDA-MB-231 was thoroughly studied, the inhibitory effects of downregulated AKT-2 on non-CSC subpopulation of epithelial breast cancer MCF7 remains unclear. 

Follow the same notion, it would be beneficial to compare the anchorage-independent growth and invasion effects of downregulated AKT-2 on CSC versus non-CSC subpopulations.    Authors can simply re-plot the data from Fig 5B and 5C to generate the information stated above.

Data presented under Table 2 was confusing.  Each data point shall be split into separate columns with clear legends.  What is rather confusing occurred in the middle column of second row with 8/8 17,79%.   What does 17,79 % mean?

Data regarding tumor formation (shown in Table 2) was rather brief, and yet, the methodology section noted a full spectrum of tumor studies, as to how animals were subjected to gross necropsy consisting of a macroscopic evaluation of all the external body orifices and the examination of the cranial, abdominal, thoracic cavities and contents.

Regarding data presented in Figure 3A, how was the time frame, as being 7 ~ 17 days, of anchorage-independent culture chosen?  What would happen to cells beyond day 17th?   Follow the same notion, have authors compared the phenotypic characteristics (like tumor formation and metastasis) between two types of mamospheres, respectively, originated from the tdTomato+ and from the tdTomato- subpopulations?

Have authors examined the effects of abrogating AKT1 and AKT3 on breast cancer, before investigating AKT-2.  Any evidence supports AKT-2 is far more important than AKT-1 and AKT-3.

Regarding Fig 7B, the order of figure legend shall be layout the same as the one in which data was plotted.

Reviewer 2 Report

The authors used MDA-MB-231 cells with ALDH1A1-tdTomato reporter vector transfection as a model to study CSC characters in. This study found high level of AKT2 expression was observed in ALDH1A1-tdTomato cells. Using knock down or inhibitor are able to reduce formation of mammosphere. I think some of the major issues need to be clearfield before this study being published.

Major:

1: The MCF-7, HCT7 and SKBR3 are not TNBC cells. The MDA-MB-468 and another TNBC should be included and compared in Figure 5 and 7. Since this study title focus on TNBS.

2: The gene enriched classification should be included in this study, instead of gene examination in figure 3D and 4A.

3: The abstract is suggested to be rewritten. For example: the meaning is confusing in page 6, line 154-158.

4: The detection quality of circulation of cancer cells need to be improved in figure 4B. It is suggest to use CELLSEARCH determination or other method to prove CTC numbers and characters.

5: The final confirmation experiment should be examined in vivo. The animal study of distance metastasis of both AKT2 silence and akt2 inhibitor should be included in this study.

6: How is the patient survival rate of AKT2 with TNBC or breast cancer.

7: The quality of western is pool and should include wild type cell in each western blot.

Minor:

1: The Image of animal model in table 2 should be illustrated.

2: The protocol of circulating tumor cells (CTC) isolation and following examination should be detailed descript.

3: How are the expressions of AKT1 and AKT3 in with/without ALDH1A1 expressed MDA-MB-231 cells.

4: The authors used with/without ALDH1A1 expressed MDA-MB-231 to find out the differences. Therefore, it is inappropriate to descript MDA-MB231 with ALDH1A1 expression as a CSC in MDA-MB231 in the beginning of abstract, page 1, line 28-29.

5: There is no the description of bar scale in figure 2.

6: Reference: page 1, line 43. Page 2, line 64

Reviewer 3 Report

Gener et al describe some novel findings in relation to TNBC and CSC characteristics in 3 different cell line models by isolating the CSCs and comparing them to non-CSCs from the same cell lines. They have also employed an in-vivo model to understand the differences and stumbled upon some exciting findings in the non-CSC model (in-vivo). This was very important as they find AKT2 inhibition in-vitro reduces the invasiveness and anchorage independent growth characteristics in all cell lines except MCF-7 where it was evident in only the CSC subpopulation. Overall, this is an interesting study and has very important findings, however it lacks execution in terms of presenting the results that undersells its importance. The authors need to rewrite it in a manner that is not so harsh on the reader. Making it easy to read will enhance the study and show its importance overall. English is written poorly and in the wrong manner in many places, this takes away the attention of the reader from the main points. There were few typos as well that I will indicate below.

Minor points-

·         Abstract- line 25-26 is very confusing and needs to be written in a concise and clear manner.

·         Introduction- The authors here mention no targeted therapy for TNBC, however they forgot to mention that patients with germline BRCA1/2 mutations have PARP inhibitors approved as a targeted therapy. Because BRCA1/2 mutations are most common in TNBC therefore this has to be mentioned in the context of TNBC as a whole although might not be relevant for their paper. They may mention if the cell lines they utilised have known BRCA1/2 mutations?

·         Line 43- 46. Needs to be rewritten differently.

·         Figures- Bar graphs in all the figures need to be same. Put significance indicators such as stars to show the significance for each comparison. It is very hard to go and read the text and figure that out- although the graphs are quite convincing but still need the indicators on them.

·         Line 135 is very poorly written and needs to be changed

·         Results- A lot of in-text description of the figures are hard to follow as to which figure is being discussed. Please have the figure numbers properly indicated, example- line 158 is very confusing to figure out which figure is being referred to and I had to read it few times to gather this information.

·         Line 156- Table 2 is missing

·         Line 160- Typo- Figure 4C instead of 5C

·         Line 170- mammosphere spelling is wrong  

·         Line 178 to 181- rewrite these sentences without grammatical errors

·         Conclusion section from this study are missing

Major points-

Pharmacological inhibition in-vitro shows efficacy of the anti-AKT2 agent. Authors indicate it is not possible currently for specific delivery of AKT2 inhibitor. However, they use transient knockdown of AKT2 and see convincing results. To complete this study, a stable knockdown of AKT2 in at least MDA-MB-231 cells and then repeating the experiment in-vivo as was done after sorting the CSCs vs Non-CSCs and injecting into mammary fat-pad is integral to this study.  Characterising the model will make it more convincing. There was a lot of focus on targeting AKT2 for TNBC however, it is loosely based on literature but evidence from their own study was lacking. It needs to be either done as suggested or provide concrete evidence/explanation as to why this was not done. Given the increased focus of targeting AKT2 in the conclusions of this study it seems a proof-of-principle in-vivo study targeting AKT2 by stable knockdown will greatly elevate the findings of this study.  

Please note that BAY 1125976 is a dual AKT1/2 inhibitor that has shown efficacy in multiple models of cancer including breast cancer. Perhaps it is worth mentioning in the discussion.  

Reviewer 4 Report

Authors have studied the role of AKT2 in triple negative breast cancer. Authors suggest that AKT2 inhibition can prevent breast cancer as inhibition leads to the reversal of Cancer stem cells mesenchymal phenotype to a more epithelial-like version along with additional reasons. I have the following comments

Major concerns:

1.       Cancer Stem Cells (CSCs) ….. progression and resistance.> this sentence in the introduction is not looking very appropriate. It looks like this is the only causative agent which may not be true.

2.       reinforce its suitability of AKT2 inhibition for cancer management.> This is a very strong statement in the abstract. Authors have made strong conclusions at other places as well. Considering the amount and volume of data authors have, the conclusions are too strong. I suggest authors diluting the claims and present it as a potential target instead of declaring AKT2 inhibition as a clinical target. Strong conclusions need more data and some clinical trial results.

3.       In vivo, an analogous high-level AKT2 is detected also in circulating tumor cells (CTC), what just further reinforce its suitability of AKT2 inhibition for cancer management.>>I feel this statement also very strong.

4.       only standard of care is chemotherapy….. since targeted therapies have failed >> last year FDA has approved the targeted therapy with olaparib.

5.       CSCs and has proved successful for the MCF7>> proved successful in what respect with respect to CSCs? Does these cell lines have these phenotypes inherently?

6.       regulated by AKT2, silencing of this oncogene should be regarded as an effective strategy >> again this statement is too strong. Anything should be done depends only on whether it is proved or not. Otherwise, it may be explored as a potential strategy.

7.       At few places, data not shown is written. There is no restriction to upload remaining data as supplementary or as a separate file to figshare. Some good example is discussed at PMID: 30793024.

8.       while just in 6 / 8 animals>> 6 out of 8 is still a good number and the median volume is larger.

9.       AKT2 thus postulates as an ideal candidate for cancer treatment.>> Current data doesn’t support this strong conclusion. Authors can present AKT2 as a good candidate needs further analysis to prove its role in cancer treatment.

10.   siAKT2 based delivery system suggested by authors for clinical use. However, siRNA based strategies have not proved an ideal way in clinic.

11. The conclusion section is empty which is mandatory.

Minor:

1)      Acknowledgment section is missing

2)      Conflict of interest: authors have declared no conflict of interest. However, mentioned about apart from those disclosed. There is no mention of what is disclosed anywhere.

Round 2

Reviewer 3 Report

Changes made have really lifted the manuscript to a higher scientific level. You might want to recheck some minor text errors and typos